# CRPGAN: Learning image-to-image translation of two unpaired images by cross-attention mechanism and parallelization strategy

**Long Feng**, **Guohua Geng***, **Qihang Li, Yi Jiang, Zhan Li, Kang Li**

School of Information Science and Technology, Northwest University, Xi'an, China

* ghgeng@nwu.edu.cn

## Abstract

Unsupervised image-to-image translation (UI2I) tasks aim to find a mapping between the source and the target domains from unpaired training data. Previous methods can not effectively capture the differences between the source and the target domain on different scales and often leads to poor quality of the generated images, noise, distortion, and other conditions that do not match human vision perception, and has high time complexity. To address this problem, we propose a multi-scale training structure and a progressive growth generator method to solve UI2I task. Our method refines the generated images from global structures to local details by adding new convolution blocks continuously and shares the network parameters in different scales and also in the same scale of network. Finally, we propose a new Cross-CBAM mechanism (CRCBAM), which uses a multi-layer spatial attention and channel attention cross structure to generate more refined style images. Experiments on our collected Opera Face, and other open datasets Summer↔Winter, Horse↔Zebra, Photo↔Van Gogh, show that the proposed algorithm is superior to other state-of-art algorithms.

**Data Availability Statement:** The Summer2Winter datasets can be obtained on website (https://people.eecs.berkeley.edu/~taesung_park/CycleGAN/datasets/summer2winter_yosemite.zip)

## Introduction

The unsupervised image-to-image translation (UI2I) is to translate an image from one domain to another, capable of changing the appearance of a given image while keeping its geometry unchanged. For example, from a horse to a zebra, from a low-resolution image to a high-resolution image, from a photograph to an art painting, and vice versa [1, 2]. UI2I has received a lot of attention due to its excellent performance in areas such as image style transfer [3–7], colourisation [8], super-resolution [9, 10], dehazing [11], denoising [12], image Synthesis [13], text-to-image Synthesis [14], image Generation [15, 16], and underwater image restoration [17].

In recent years, with the emergence of Generative Adversarial Networks (GANs) [18], many works with GAN has been proposed to solve the UI2I tasks [19–24]. In UI2I tasks without paired training data, the main problem of GANs is that the adversarial loss [18] is un-constrained and many mappings functions exist between the source and target domains, which

The Horse2Zebra datasets can be obtained on website (https://people.eecs.berkeley.edu/~taesung_park/CycleGAN/datasets/horse2zera.zip) The Photo2Van Gogh datasets can be obtained on website (https://people.eecs.berkeley.edu/~taesung_park/CycleGAN/datasets/vangogh2photo.zip) The Grumpifycat datasets can be obtained on website (https://people.eecs.berkeley.edu/~taesung_park/CycleGAN/datasets/grumpifycat.zip).

**Funding:** This research was funded by the National Key Research and Development Program of China (2020YFC1523301 and 2019YFC1521103), National Natural Science Foundation of China (62271393), Key Research and Development Program of Shaanxi Province (2019ZDLSF07-02, 2019ZDLGY10-01 and 2021GY-171), National Natural Science Foundation of China (61731015), Key Research and Development Program of Qinghai Province (2020-SF-142). The funders had no role in study design, data collection and analysis, decision to publish, or preparation of the manuscript.

**Competing interests:** The authors have declared that no competing interests exist.

may lead to unstable training and failure of image translation. To solve this problem, Cycle-GAN [3], DiscoGAN [25] and DualGAN [26], introduce the cycle-consistency loss [3] to network model and learn the reverse mapping from target to the source domain with the reconstruction consistency constraint.

The above methods usually require a large number of unpaired images for training. However, the massive unpaired images are difficult to be obtained. Therefore, the Few-Shot and One-Shot learning has attracted more and more reserchers' interest [27–29]. In One-Shot unsupervised learning, the source and target domain each has only one image and these two images are unpaired. Unfortunately, one-shot and few-shot usually leads to severe overfitting of the model. Therefore how to solve UI2I task with a small number of training samples faces great challenges.

The recently proposed SinGAN [30] shows that there are enough information contained in the patches of one image to train a GAN model, thus a large amount of information can be used to extracte from a single image. Unfortunately, the SinGAN is limited to generate a specific data distribution, not suitable for UI2I task.

Furthermore, due to lack of constraints, SinGAN is a serial multi-level model structure with slow training and is limited to generating specific data distributions, resulting in blurred image translated images translation results. ConsinGAN [31] uses parallelism for the first time to improve the training speed, but still does not solve the image translation blur problem of UI2I. TuiGAN [32] takes full advantage of SinGAN's learning of translated images by multiple scales and using consistency loss to limit the structural difference between two images, which can achieve UI2I of two unpaired images.

However, as with SinGAN, it is limited by the serial structure of the model resulting in slow training, and cannot effectively capture the differences between the source and target domains at different scales due to the constant changing of the perceptual field to extract the underlying relationship between the two images. Therefore, this learning process can generate a large amount of noise, resulting in poor image translation quality, distortion and others that do not accord with human vision.

To overcome the above problems, we propose a new one-shot image translation network framework, named CRPGAN, which adopts multi-scale training structures and progressive growth generators that can effectively capture the differences in distribution between the source and target domains. In the multi-scale learning process, each scale has two generators to generate target image and restore the source image, and one discriminator to capture the domain distribution of the source and target domains. We divide the initial generator structure into three convolutional blocks, H, B, and T, and keep adding new B block while keeping H and T blocks constantly in high-scale training to continuously optimize the details of image translation. That is, a new B block is added to the generator of scale $N-1$ to form the generator of scale $N$. In addtion, to improve the training speed, we make use of parallel structures that do not require repeated training of the current generator. Finally, we use two parameter sharing structures to reduce the training time, the first sharing is to share the generator network parameters between different scales, and the second is to share parameters between convolutional blocks B in the same scale generator. Moreover, in order to avoid overfitting, a gradually decreasing learning rate is applied to the additional B blocks, which can more accurately capture the different domain distributions at each scale.

During multi-scale training, the problem of excessive noise and blurred image translation results can easily occur due to repetitive training of single sample image. However, CBAM [33] is often effective in many tasks such as object detection [34], image classification [35], image translation, etc., capturing local information from the image in terms of channels and space,

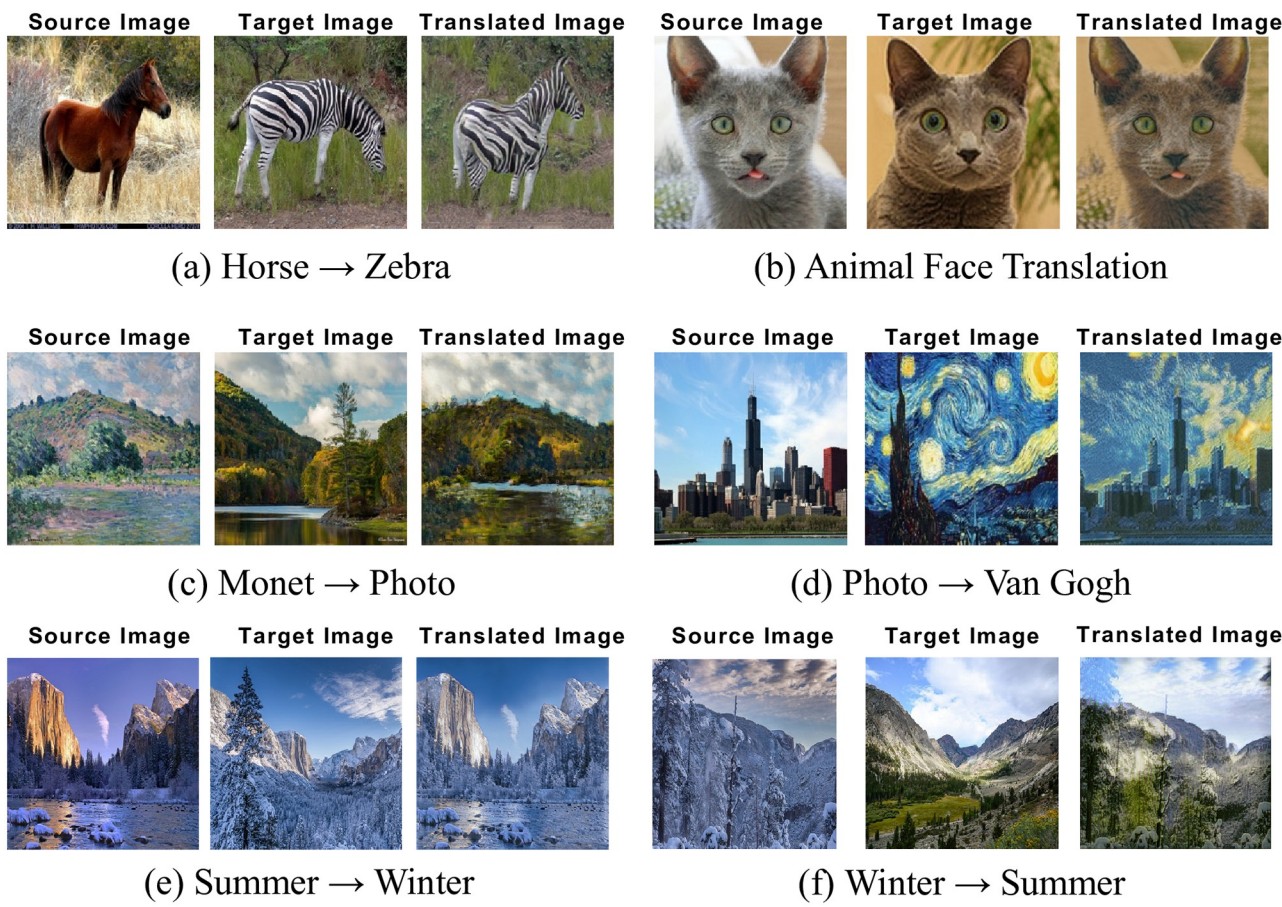

**Fig 1. Several results of our proposed method on various tasks.** Here we show object transformation ((a), (b)) and image style transfer((c)-(f)). The three images from left to right are the source image (which provides the main content), the target image (which provides style and high-level semantic information), and the translated image.

thus improving image quality. We find that adding CBAM to the one-shot image translation module still produced images with a lot of noise.

To address this problem, we propose a novel Cross-CBAM mechanism (CRCBAM), which uses a two-layer spatial attention and channel attention crossover structure to fully learn local and global information by learning the semantics and location of the target style multiple times in space and channel, in order to better capture differences between sources and target domains at different scales in order to generate more finely stylised images.

Our CRPGAN can significantly reduce the training time (60 minutes versus 240-300 minutes of TuiGAN [32]). We have conducted extensive experiments on various UI2I tasks including Summer→Winter, Horse→Zebra, Photo→Van Gogh and so on. The results show that our method can effectively solves the problem of low translation quality in one-shot learning. Compared with the existing UI2I models, CRPGAN achieves better performance. Some experimental results of our method are shown in Fig 1.

The main contributions of this paper are as follows:

- We propose a new one-shot image translation network framework CRPGAN, which introduces a progressive growth generator and a multi-scale training approach to efficiently

capture the differences between two different distributions in the source and target domains to learn image target styles from global information.

- We use a parallel structure and a two-layer parameter sharing mechanism to reduce the training time for one-shot image translation.

- We propose a Cross-CBAM attention mechanism(CRCBAM) to fully learn the local information of single-sample image to generate finer stylized images.

The paper is structured as follows. Section 1 is a general introduction providing the necessary background and problems of unsupervised image translation. Section 2 provides current work related to style transfer, unsupervised image translation, and one-shot image translation. Section 3 provides the network structure, attention mechanism and loss function of the methods in this paper. Section 4 provides experimental setup, baseline methodology and evaluation metrics. Section 5 provides qualitative and quantitative experiments and the corresponding parametric and ablation studies. Section 6 provides a conclusion of our approach and future work.

## Related work

### Image style transfer

Image style transfer [1, 2, 36–38] aims to transfer the style of target image to the source image. The early works on image style transfer include [1, 2]. Commonly, these methods are based on convolutional neural network and transfer image by minimizing the Gram matrix of pre-trained deep features. [1] combines the content and style of a image to transfer the image style. However, the cost of its training process is very high. Considering the correlation matrix between the deep features extracted by neural network is helpful for image transfer, the [2], which minimizes the loss constructed by the Gram matrix or covariance matrix, captures the visual style of target image to transfer image.

[36, 39, 40] attempt to transfer style via a single network layer in a trained feed-forward neural networks. The shortcoming of these feed-forward methods is that each network is limited to one image style, the optimization is slow and the model is not flexible enough. The work of Ioffe and Szegedy [41] introduces a batch normalisation (BN) layer, which significantly simplifes the training of feedforward networks by normalizing the statistical information of the features.

### Image-to-image translation

The work of Pix2Pix [42] achieves impressive results in image-to-image translation tasks using paired images based on conditional generative adversarial networks (CGAN), which has been extended to other tasks such as super-resolution [9, 10] and video generation [43, 44] etc. Although these methods achieve good results, all of them need to collect pair of data for training, which limits their practical application. Unlike Pix2Pix [42], CycleGAN [3] and Disco-GAN [25], DualGAN [26] solved unsupervised image-to-image translation (UI2I) by introducing one cycle-consistency loss and two opposite domain transformation generators. Liu [27] further extends the idea of cyclic consistency and proposed a FUNIT for few-shot UI2I, which replace the domain-specific space with a domain-shared potential space. DRIT [5] embeds the images into a domain-invariant and a domain-specific style space to translate image. But, the above UI2I methods not only require large amounts of data and computation resources, but also cannot effectively capture the differences of the distribution between the source and target domain.

Benaim [29] and Cohen [45] propose two methods to solve the problem of One-Shot cross-domain translation by learning the unidirectional mapping between the source domain with one image and the target domain with a set of images. By adopting the generative model and a multi-scale structure, Lin [32] used a bidirectional function to map image from source to target and then map back to source domain to solved the One-Shot cross-domain translation task. The disadvantage of these methods is that they eithor need a large number of training images, or has a high time complexity and will lead to low image translation quality, distortion and other conditions that do not match human vision perception.

## One-shot image-to-image translation

Shaham T R et al. proposed SinGAN [30] using an unconditional pyramid generation model to learn patch distributions based on images of different scales for one-shot image translation. ConSinGAN [31]adopts the multi-level parallel training method to improve the speed of model training, but it can not capture the domain distribution between images or solve image blur problem. Recently, Lin et al proposed TuiGAN [32] continuing to capture the domain distribution between two images using a multi-level approach to one-shot cross-domain translation. This approach constantly changes the receiving domain to extract the potential relationship between the two images and cannot effectively capture the differences between source and target domains at different scales. Therefore, this learning process produce a lot of noise, resulting in poor image translation quality, distortion and other phenomena that do not conform to human vision. And it takes 4-5 GPU hours, which does not allow for better control of image translation results.

In this paper, in order to generate high-quality the UI2I translation image quickly, we propose a parallel multi-scale structural training and progressive growth generator to obtain mapping function between two unpaired images. This method overcomes the shortcomings of the existing UI2I methods, such as large data sets, large memory resources, low translation quality and long training time.

## Methods

Our method is to learn the mapping function from source domain A to target domain B through parallel multi-scale training and progressive growth generators, where A and B denote two image domains, e.g. summer and winter.

We summarize our approach from four aspects. 1) We replace the multi-scale serial network model proposed by TuiGAN [32] with parallel multi-scale network model, which need not to use the generated images in the previous scale and share the network parmaters between the adjacent scales and also in the same scale. This parallel multi-scale network model can train independently without affecting the image transfer performance and greatly improve the training speed. 2) We adopt a progressive generator to continuously add new convolutional layers during the training process, which can capture the differences in domain distribution at different scales and obtain the detailed features of image translation. 3) We use two network parameter sharing mechanism to speed the training. The first is to share the network parameters in the previous $N - 1$ and current scale $N$, and the newly added layers and the kept in B blocks. 4) We use the newly proposed CRCBAM(Cross-CBAM) attention mechanism to fully extract the image local information to generate a finer stylized image. Moreover, to avoid training overfitting, a gradually decreasing learning rate is used to capture different domain distributions more accurately in different scales.

Unlike traditional UI2I methods, our method requires only two unpaired images to complete the UI2I task, which can produce high quality translation images with fast training speed. Below, we provide a detailed description of the our proposed network.

## A, Network architecture

The network architecture of the proposed CRPGAN is shown in Fig 2. The overall framework consists of two symmetric pyramids generators ($G$ and $F$) and discriminators ($D_A$ and $D_B$). It consists of two main translation modules: generator G for translating $I_A$ into $I_{AB}$ (Fig 2(a)) and generator F for translating $I_B$ into $I_{BA}$ (Fig 2(b)), $D_A$ for distinguishing whether the input image is from the real image $I_A$ or the generated image $I_{BA}$, and $D_B$ for distinguishing whether the input image is from the real image B or the generated image $I_{AB}$. In Fig 2(a), generator G translates $I_A$ into $I_{AB}$, followed by generator F reconstructs the translation result $I_{AB}$ into $I_{ABA}$. In Fig 2(b), generator F translates $I_B$ into $I_{BA}$, followed by generator G reconstructs the translation result $I_{BA}$ into $I_{BAB}$. Generators G and F have the same network structure and form a symmetric structure in the network with different weight parameters. The learning model constains four losses: $L_{ADV}$, $L_{CYC}$, $L_{Content}$, $L_{TV}$, in which the adversarial loss $L_{ADV}$ ensure the generated images are similar to the target images, the cyclic consistency loss $L_{CYC}$ solves the collapse problem in GAN, the content loss $L_{Content}$ can maintain the content information of source image, and the total variance loss $L_{TV}$ can avoid the noise and artifacts occured in the translated image.

Since there are only two images ($I_A$ and $I_B$) in the training sample, in order to make fully use these two images and extract as much image information as possible, we adopt multi-scale structure, which takes different resolutions of image as inputs of our model. The whole network is divided into $N$ scales, namely Scale 0, Scale 1, ..., Scale $N$, which is shown in Fig 2. We downsample $I_A$ and $I_B$ to $N$ different scales, $\mathcal{I}_A = \{I_A^n \mid n = 0, 1, \cdots, N\}$ and $\mathcal{I}_B = \{I_B^n \mid n = 0, 1, \cdots, N\}$, where $I_A^n$ and $I_B^n$ are downsampled from $I_A$ and $I_B$ by a scale factor $(1/s)^n (s \in \mathbb{R})$, respectively.

To learn the mapping between the source and target domains at different scales, we introduced progressively growing generators into the network. At Scale 0, our initialised generator consists of Head block (H), Body block (B) and Tail block (T), where the Head block network structure is $Conv - BN - ReLU$, the Body block network structure is composed of three $Conv - BN - ReLU$ and CRCBAM attention mechanisms, and the Tail block network structure is $Conv - Tanh$. At the next scale, our generator keeps the number of Head blocks (H) and Tail blocks (T) constant and dynamically adds a Body block (B) to fully learn the image local features.

Instead, we used a parallel multi-scale training structure to speed up the training process without using the image translation results from the previous scale, i.e. without using the translated image of the previous scale $N - 1$ as input for the current scale $N$, structurally speeding up the training speed.

$$I_{AB}^n = G^n(I_A^n), \quad I_{BA}^n = F^n(I_B^n) \tag{1}$$

Secondly, due to the similar structure of the multi-scale generators used in our method, repeated initialization of generator training will increase the training time cost. Therefore, we uses parameter sharing to assign the weights of the previous scale training parameters to the weights of the current scale training parameters to speed up the training, which means that $G^{n-1}$ and $F^{n-1}$ in equation Eq (2) do not need to be initialised and directly use the weight parameters of the previous scale training, and only need to initialise the newly added Body

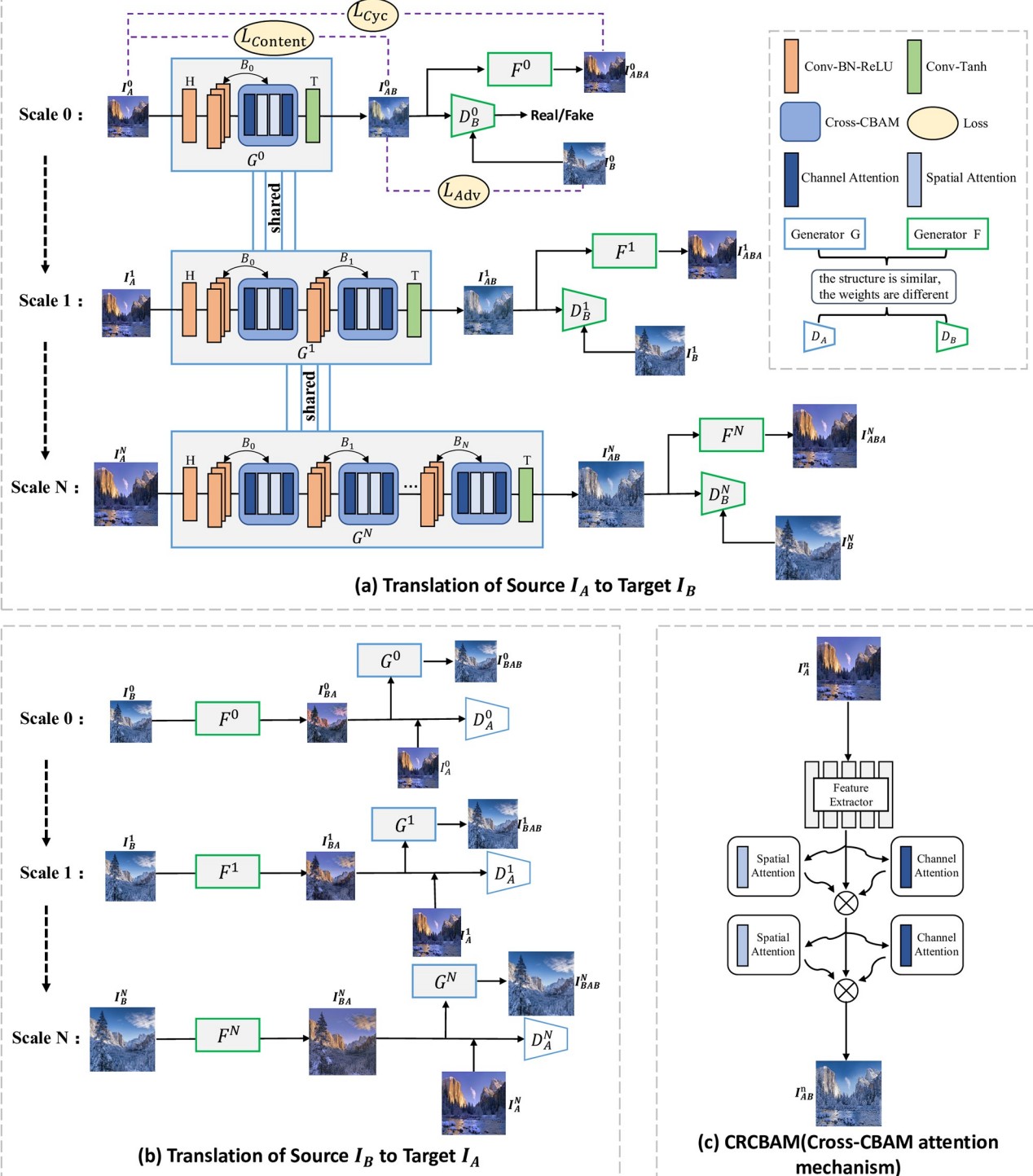

**Fig 2. CRPGAN network architecture.** In (a) and (b), our model consists of two symmetric pyramids of GANs that gradually refine the generated images from global structure to local details. We start training at 'scale 0' by using the lowest resolution image and the smallest generator. With the scale increasing, the size of the generator are gradually increased and the resolution of the image are also changed from low to high. (c) is our Cross-CBAM attention mechanism to extract the image global information and local information.

block $B_{new}$ parameters.

$$G^n = G^{n-1} + B_{new}, \quad F^n = F^{n-1} + B_{new} \tag{2}$$

where $G^n$ is a generator from source domain A to target domain B at scale n, and $F^n$ is a generator from source domain B to target domain A at scale $n$. $G^{n-1}$ is a generator from source domain A to target domain B at scale $n-1$, and $F^{n-1}$ is a generator from source domain B to target domain A at scale $n-1$. $B_{new}$ is the newly added Body block.

In addition, since adding Body blocks with the same initialisation weight parameters each time the network structure is added will likewise increase the training time, we continue to use parameter sharing so that the parameters of the newly added Body block $B_{new}$ at the current stage are the same as those of the adjacent $B_{n-2}$, again increasing the training speed.

$$
\begin{aligned}
G^{n-1} &= H + B_0 + B_1 + \cdots + B_{n-2} + T, \\
G^n &= H + B_0 + B_1 + \cdots + B_{n-2} + B_{\text{new}} + T, \\
F^{n-1} &= H + B_0 + B_1 + \cdots + B_{n-2} + T, \\
F^n &= H + B_0 + B_1 + \cdots + B_{n-2} + B_{\text{new}} + T
\end{aligned}
\tag{3}
$$

where H is a Head block, T is a Tail block, $B_0, B_1, \ldots, B_{n-2}$ are Body blocks already trained by generator $G^{n-1}, F^{n-1}$, and $B_{\text{new}}$ is a new Body block added to $G^{n-1}, F^{n-1}$ to form the new generators $G^n$ and $F^n$.

## B, Cross-CBAM mechanism

The traditional CBAM attention mechanism has designed to infer attention weights in both channel and space dimensions, multiply them with the original feature map to adaptively adjust features, and improve accuracy and reduce training costs in many tasks (classification, object detection, etc.).

As shown in Fig 3(a), given an intermediate feature map $F \in \mathbb{R}^{C \times H \times W}$ as the input of channel attention $M_c$ and spatial attention $M_s$. The channel attention $M_c$ and spatial attention

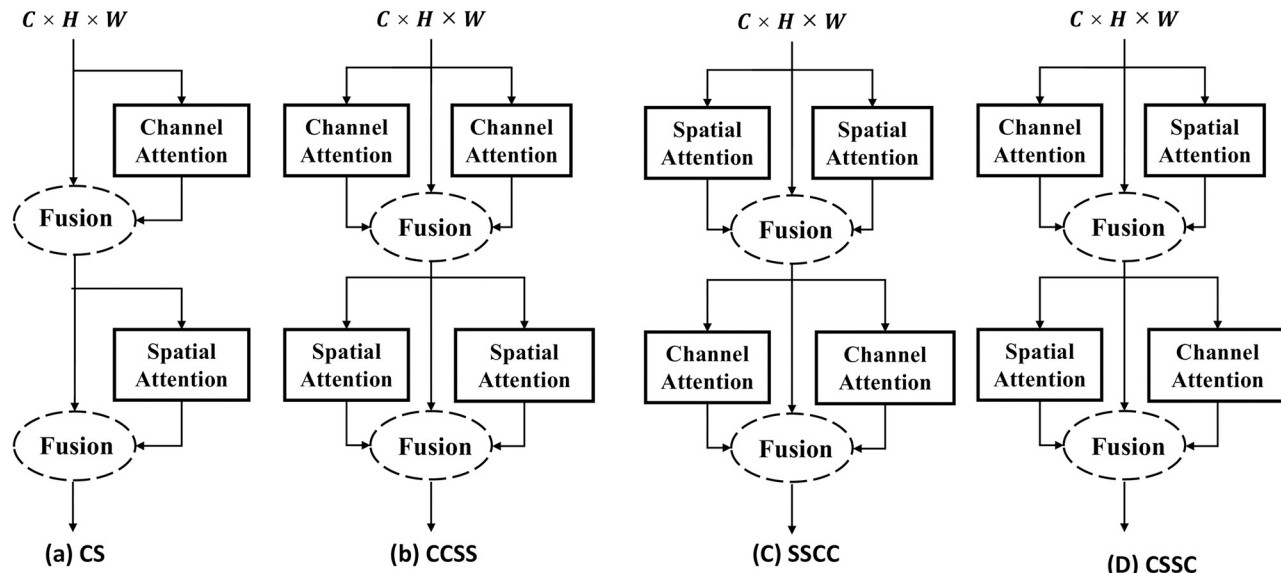

**Fig 3. Structure comparison of the different order of channel and spatial attention.** (a) is CBAM attention mechanism including a channel attention and spatial attention mechanism. (b) is the mechanism of two channel followed by two spatial attentions. (c) is the mechanism of two spatial attentions followed by two channel attentions. (d) is our CRCBAM attention mechanism.

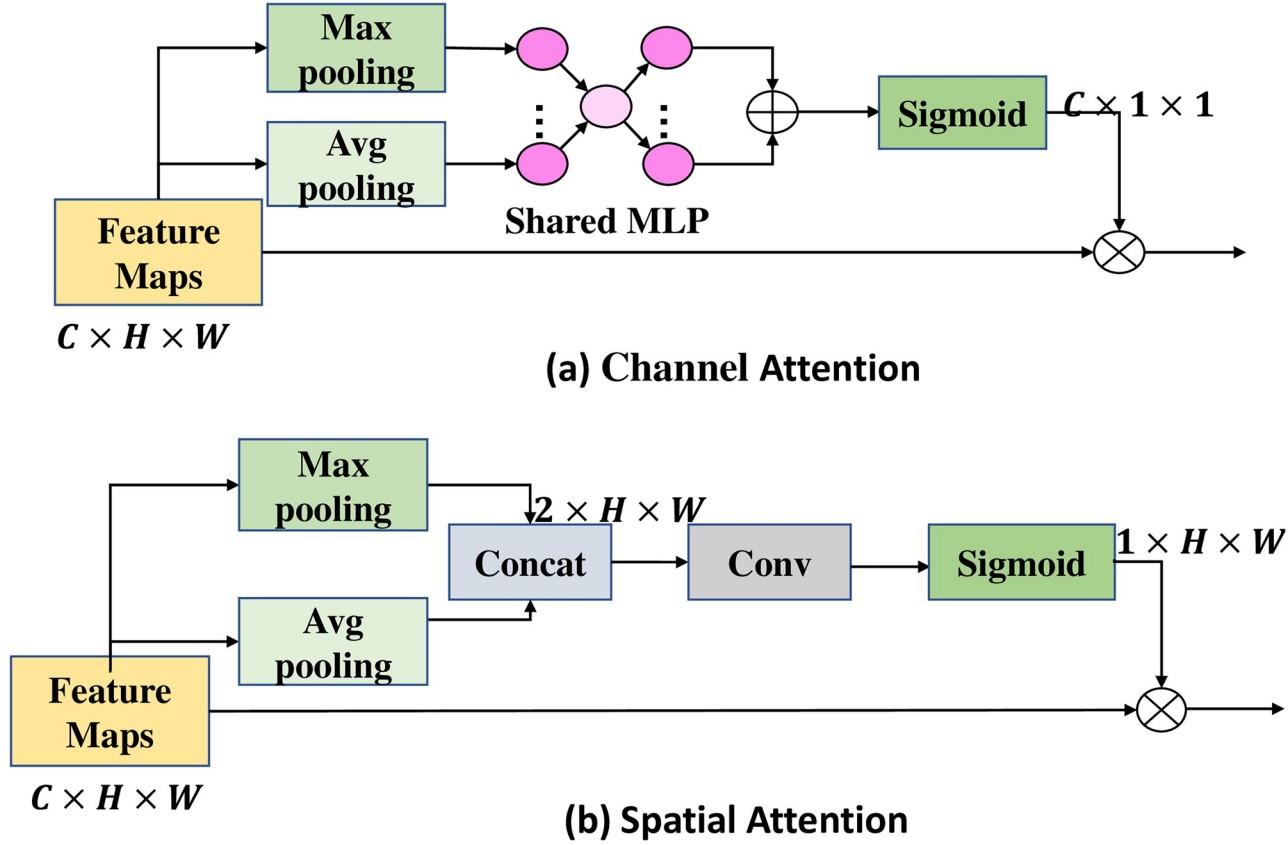

**Fig 4. Channel attention and spatial attention mechanism of CBAM.**

mechanisms $M_s$ are in Fig 4. $M_c \in \mathbb{R}^{C \times 1 \times 1}$ denotes the 1D channel attention map, $M_s \in \mathbb{R}^{1 \times H \times W}$ denotes the 2D spatial feature map.

$$F' = M_c(F) \otimes F,$$
$$F'' = M_c(F') \otimes F' \tag{4}$$

where $\otimes$ denotes element-wise multiplication, $F'$ is the feature map after the channel attention calculation, $F''$ is the final refined output.

The above is the CBAM process, these results are related to the task of large data sets, but in the one-shot learning process, it is impossible to better learn semantic information and location information, local information and global information are seriously missing, so the one-shot image translation task generates significant noise (Fig 5(a)).

To address this, we hypothesize the use multiple crossed channels and spatial attention to address this shortcoming, and design three strategies, such as Fig 3(b)–3(d). (b) uses the fused results of features obtained from two parallel channel attentions as the input of two parallel spatial attentions, and multiplies the results with the original feature map to adaptively adjust the features, (c) uses the fused results of features obtained from two parallel spatial attentions as the input of two parallel channel attentions, and multiplies the results with the original feature map to adaptively adjust the features, (d) The result of feature fusion using parallel channels and spatial attention intersection structures is used as input to the cross structure, and finally the result is multiplied by the original feature map to adaptively adjust the features.

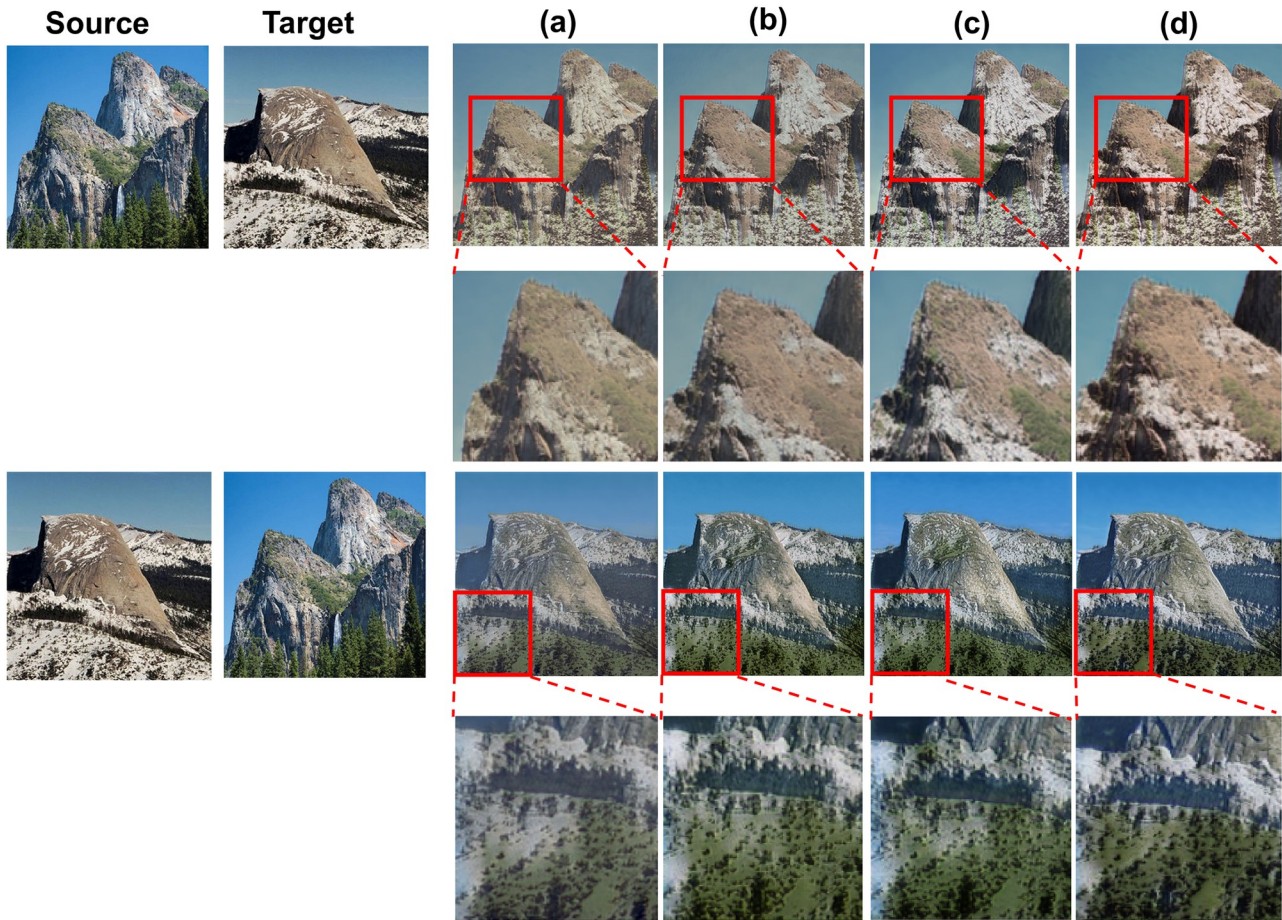

**Fig 5. The visualization results of the different order of channel and spatial attention.**

Fig 3(d) is the Cross-CBAM attention mechanism approach in this paper, through the cross-structure of two channels and spatial attention mechanism to learn the semantic information and position information of single image from the channel and spatial dimensions multiple times, to optimize the local information of single-sample image translation.

In Fig 3(d), our overall attention process can be summerized as:

$$F' = M_c(F) \otimes F \oplus M_s(F) \otimes F,$$
$$F'' = M_c(F') \otimes F' \oplus M_s(F') \otimes F' \tag{5}$$

where $\otimes$ denotes element-wise multiplication, $\oplus$ denotes element-wise summation. $F''$ is the final refined output. $M_c$ is the channel attention mechanism, $M_s$ is the spatial attention mechanism.

The results of applying a personal attention strategy to a single image are shown in Fig 5 and Table 1. (a) is the stylized image adding CBAM, we find the generated image is blurred. In (b) and (c), we find that the clarity and style of image translation results gradually improved, but local information remained relatively blurred. (d) Because of our attention mechanism, we find that the content and style of image translation results are relatively clear, with the lowest SIFID values.

**Table 1. Quantitative comparison between different order of channel and spatial attention $\eta$ of CRPGAN in terms of SIFID.** The best scores are in bold.

| Metrics | CRPGAN (SIFID↓) | | | |
|---|---|---|---|---|
| | (a) | (b) | (c) | (d) |
| Summer→Winter | 1.035 | 0.887 | 0.390 | **0.385** |
| Winter→Summer | 1.642 | 0.528 | 0.523 | **0.496** |

## C, Loss functions

CRPGAN comtains four loss functions, including adversarial loss, cycle-consistency loss, content loss, and total variation loss. The details are described below.

**(1) Total loss.** For any $n \in \{0, 1, \cdots, N\}$, the overall loss function of the $n$-th scale is defined as follows:

$$\mathcal{L}^n_{\text{ALL}} = \mathcal{L}^n_{\text{ADV}} + \lambda_{\text{CYC}}\mathcal{L}^n_{\text{CYC}} + \lambda_{\text{Content}}\mathcal{L}^n_{\text{Content}} + \lambda_{\text{TV}}\mathcal{L}^n_{\text{TV}} \qquad (6)$$

where $\mathcal{L}^n_{\text{ADV}}, \mathcal{L}^n_{\text{CYC}}, \mathcal{L}^n_{\text{IDT}}, \mathcal{L}^n_{\text{Content}}$ and $\mathcal{L}^n_{\text{TV}}$ refer to adversarial loss, cycle-consistency loss, identity loss, content loss, and total variance loss at the $n$-th scale respectively. $\lambda_{\text{CYC}}, \lambda_{\text{IDT}}, \lambda_{\text{Content}}$ and $\lambda_{\text{TV}}$ are the hyperparameters that balance the loss functions.

**(2) Adversarial loss.** We use adversarial loss to encourage the generator to transfer images that are visually similar to the target domain images. At each scale of image translation, there are two discriminators, $D^n_A$ and $D^n_B$, with $I^n_A$ and $I^n_B$ as inputs, respectively, and output is the probability that the input is a natural image in the corresponding domain. In this paper, we choose WGAN-GP loss [46] as adversarial loss, which can effectively increase the stability of training through weight clipping and gradient penalty.

$$\begin{aligned} \mathcal{L}^n_{\text{ADV}} = {} & D^n_B(I^n_B) - D^n_B(G^n_{AB}(I^n_A)) \\ & + D^n_A(I^n_A) - D^n_A(G^n_{BA}(I^n_B)) \\ & - \lambda_{\text{PEN}}(\|\nabla_{\hat{I}^n_B} D^n_B(\hat{I}^n_B)\|_2 - 1)^2 \\ & - \lambda_{\text{PEN}}(\|\nabla_{\hat{I}^n_A} D^n_A(\hat{I}^n_A)\|_2 - 1)^2 \end{aligned} \qquad (7)$$

where $\hat{I}^n_B = \alpha I^n_B + (1-\alpha)I^n_{AB}, \hat{I}^n_A = \alpha I^n_A + (1-\alpha)I^n_{BA}, \alpha \sim U(0, 1)$ and $\lambda_{\text{PEN}}$ is the penalty coefficient.

**(3) Cycle-consistency loss.** One of the training problems of conditional GAN is mode collapse. To mitigate the mode collapse problem, we impose cycle-consistency loss [3] on the generator, which can constrain the model to retain the inherent properties of input image after translation: $\forall n \in \{0, 1, \cdots, N\}$,

$$\mathcal{L}^n_{\text{CYC}} = \|I^n_A - I^n_{ABA}\|_1 + \|I^n_B - I^n_{BAB}\|_1 \qquad (8)$$

where $I^n_{ABA} = F^n(I^n_{AB}), I^n_{BAB} = G^n(I^n_{BA})$.

**(4) Content loss.** To maintain the content information of input image, we include the content loss $L_{CONTENT}$ with calculating the mean-square error between the features of content and output images extracted from the pre-trained VGG-16 networks similar to the existing work

by Gatys et al [2].

$$L_{Content}^n = \sum_i \left(\frac{1}{C_i H_i W_i}\|\phi_i(I_{AB}^n) - \phi_i(I_A^n)\| \right.$$
$$\left. + \frac{1}{C_i H_i W_i}\|\phi_i(I_{BA}^n) - \phi_i(I_B^n)\|\right) \tag{9}$$

where $I_{AB}^n = G^n(I_A^n), I_{BA}^n = F^n(I_B^n)$. $\phi_i$ is the pretrain model of VGG16.

**(5) Total variation loss**. To avoid the effect of noise on the image, we introduce the total variation (TV) loss [47], which helps to remove the rough texture in the translated image to smooth the image, eliminate noise, induce spatial continuity in the translated image, and avoid over-pixelation of the result. It encourages images to consist of several patches by calculating the differences of neighboring pixel values in the image. Let $x[i, j]$ denote the pixel of image x located in the $i$-th row and $j$-th column, the $n$-th stage TV loss is calculated as follows.

$$\mathcal{L}_{TV}^n = L_{tv}(I_{AB}^n) + L_{tv}(I_{BA}^n) \tag{10}$$

where $L_{tv}(x) = \sum_{i,j}\sqrt{(x[i, j+1] - x[i, j])^2 + (x[i+1, j] - x[i, j])^2}, x \in \{I_{AB}^n, I_{BA}^n\}$.

## Experiments

Our training details, dataset, evaluation metrics, and all baselines are described below.

### A, Training details

We train the network using the Adam [48] optimizer, where $\beta 1 = 0.5$ and $\beta 2 = 0.999$. The initial learning rate $\delta$ of CRPGAN is 0.0005, the scale factor $\eta$ is 0.1, our model contains 5 scales with 100 epochs per scale, and the generator learning rate decays exponentially. In addition, we adopt the generator based on Resnet [49] and discriminator based on PatchGAN [50]. We set the batch size to 1, the maximum image resolution to $250 \times 250$, and the minimum resolution to $100 \times 100$. All experiments are set with the weight parameters $\lambda_{CYC} = 1$, $\lambda_{CONTENT} = 0.08$, $\lambda_{TV} = 0.1$ and $\lambda_{PEN} = 0.1$. We train our model by using a single 2080Ti GPU and the training costs 60 minutes.

As mentioned before, all generators in the CRPGAN framework share the same architecture and they are all fully convolutional networks. In detail, the generator consist of 5 conv-blocks in the form of 3x3 Conv-BatchNorm-LeakyReLU with stride 1. Whenever any scale $N - 1$ converges, we add three convolution layers to body block of generator. For each discriminator, we choose the Markovian discriminator [50] which have the same receptive field as the generator, and the patch size is $11 \times 11$.

### B, BaseLines

In this paper, the proposed methods are compared qualitatively and quantitatively with the latest UI2I methods, we choose the following baselines:

- SinGAN [30], which is a pyramidal unconditional generative model trained on only one image from the target domain.

- TuiGAN [32], which Uses two unpaired images for image translation by multi-stage training structure.

- CycleGAN [3], which introduces cycle-consistency loss to learn the reverse mapping from the target domain to the source domain.

- TSIT [52], which provide a carefully designed two-stream generative model for image translation.

- DCLGAN [19], which bases on contrastive learning and a dual learning setting to infer an efficient mapping between unpaired data.

- lrwGAN [23], which solves the image translation task between two unaligned domains by importance re-weighted image selection.

- StyTR2 [53], which is a style transfer model using transformers as encoders.

- Qs-Attn [24], which design a query-selected attention (QS-Attn) module to ensure that the source image learns the target image features at the corresponding location for image translation.

  For all the above baselines, we use their official released code to produce the results.

## C, Evaluation

**Metrics**. In this paper, we use Single Image Fréchet Inception Distance (SIFID) [30] to evaluate the quality of translated images. SIFID estimates the difference in the internal distribution of two images by calculating the Fréchet Inception Distance (FID) [51] between the depth features of the two images. A lower FID indicates a smaller Fréchet distance between the real image and the generated image. That is, a lower FID means that the translated image is more realistic. Therefore, a lower SIFID score means that the style of the two images is more similar and the quality of the translated image is higher. In this paper, the SIFID between the translated image and the target image is calculated.

## Results

In this section, we compare CRPGAN with all baselines on different datasets. In addition, we only use the SIFID score as an evaluation metric.

### A, General UI2I tasks

Table 2 shows the translation results of CRPGAN compared with the latest UI2I models on Summer↔Winter, Horse↔Zebra and Photo→Van Gogh. Clearly, our method outperforms all the baselines. The corresponding qualitative results for the random selection are given in Figs 6–8.

**Table 2. Average SIFID and training time of various baselines versus our method on general UI2I tasks.**

| Method | Summer↔Winter | Horse↔Zebra | | Photo→Van Gogh |
|---|---|---|---|---|
| | SIFID ↓ | SIFID ↓ | Runtime ↓ | SIFID ↓ |
| SinGAN [30] | 1.817 | 3.194 | 180mins | 1.406 |
| TuiGAN [32] | 0.768 | 2.211 | 240mins | 2.433 |
| CycleGAN [3] | 0.991 | 1.164 | 20hours | 1.023 |
| TSIT [52] | 0.905 | 5.833 | 220mins | 2.251 |
| DCLGAN [19] | 0.914 | 1.106 | 48hours | 3.374 |
| lrwGAN [23] | - | 3.532 | 48hours | - |
| StyTR2 [53] | 2.564 | 3.803 | 220mins | 1.198 |
| Qs-Attn [24] | 1.582 | 1.183 | 24hours | 4.635 |
| CRPGAN(ours) | **0.533** | **1.057** | **60mins** | **0.735** |

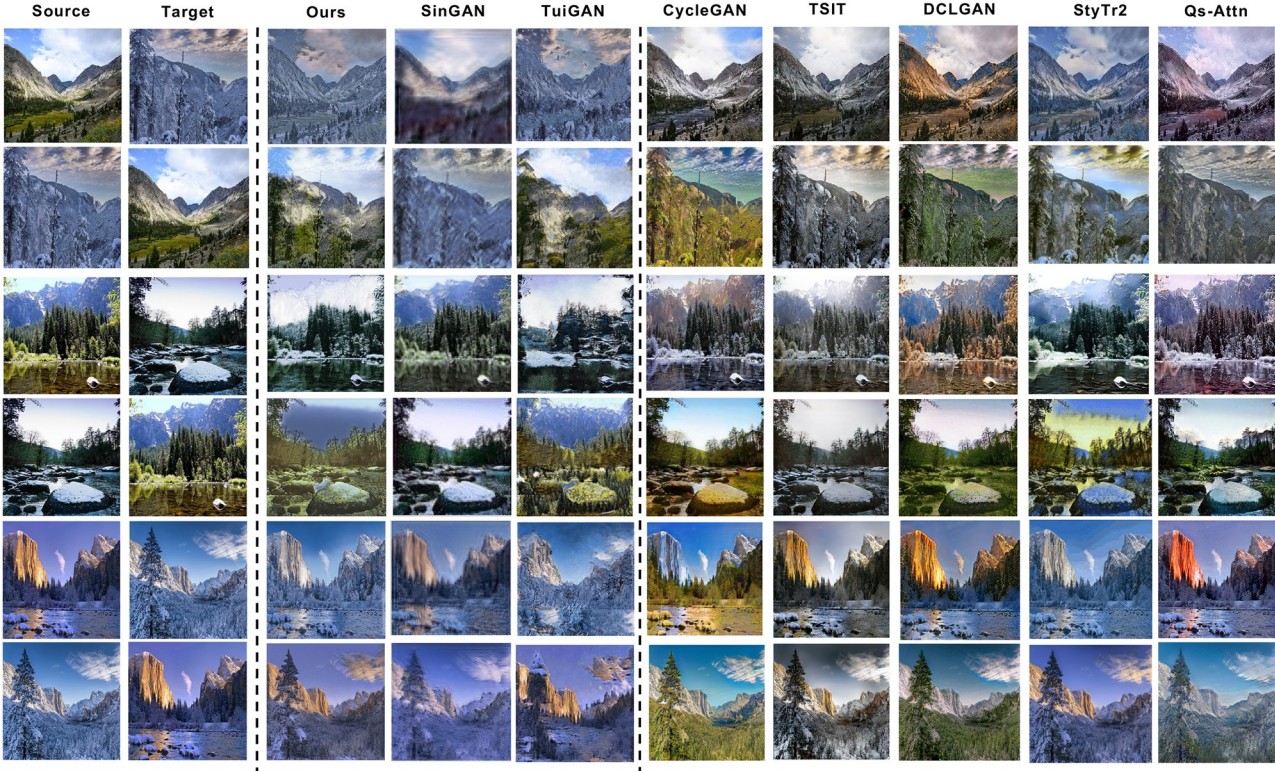

**Fig 6. Translation results of CRPGAN with various baselines on Summer↔Winter.** Among them, SinGAN is trained using one target domain image, TuiGAN and CRPGAN in this paper are trained using two unpaired images, others are trained using the complete dataset.

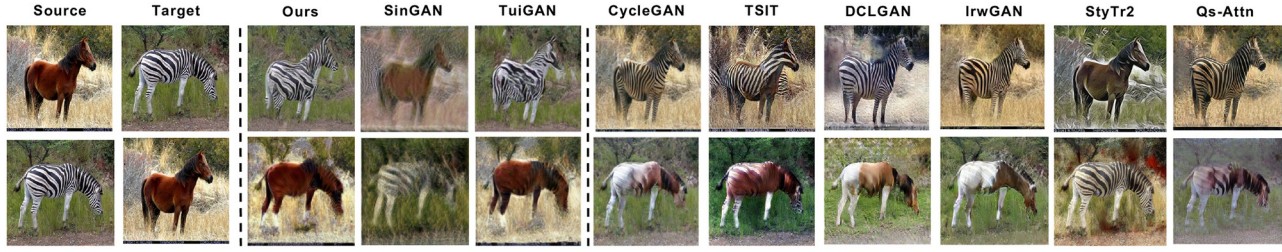

**Fig 7. Translation results of CRPGAN with various baselines on Horse↔Zebra.**

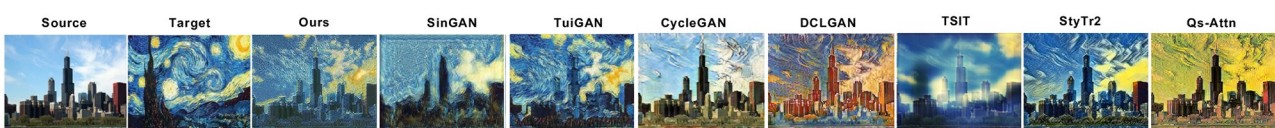

**Fig 8. Translation results of CRPGAN with various baselines on Photo→Van Gogh.**

Overall, CRPGAN generates better translation images than baselines model, with a substantial reduction in training time. SinGAN [30] change the global color of the source image in their translation results, but they fail to transfer high-level semantic structures, when the difference between image blocks is large. SinGAN is unable to learn better image distributions and

are prone to generate unrealistic images, Horse↔Zebra in Fig 7. TuiGAN [32] trained with only two images achieved comparable results, but the translated images were usually unclear and of poor quality, with noise, distortion, and other parts that do not match human vision, Summer↔Winter in Fig 6 and Horse↔Zebra in Fig 7. TSIT [52] only changes the color of the source image and does not capture the prominent painting style of the target domain. Cycle-GAN [3], DCLGAN [19] and Qs-Attn [24] can perseve source content features well, but cannot learn one-shot target image styles. StyTr2 [53] can change the image style and the source image content is well maintained, but its training time cost is too high due to its utilization of transformer structure, Photo→Van Gogh in Fig 8. CRPGAN learns the global and local structure of the source image through CRCBAM attention from coarse to fine by a multi-scale progressive generation structure, so it can preserve the architectural contours more better and has the style of the target image.

## B, Painting-to-image translation

The painting-to-image translation task is to convert a rough drawing into a realistic image. In this paper, two samples provided by SinGAN [30] are used for training, and the results are shown in Fig 9. Although the two images have similar elements trees and roads, the image styles are completely different. SinGAN and TSIT [52] are unable to transfer the target style or generate specific details leaves on trees. StyTr2 [53] is good at global style transfer, but local information (e.g. trees) does not transform. TuiGAN [32] is able to transfer the target style but the local details are not as rich as our model, as shown in the second row in Fig 9.

## C, Parametric study

To evaluate the effect of the scaling factor $\eta$ in the generator and the Head, Body, and Tail blocks on the image translation results in the network framework of this paper, we designed parameter study experiments on Horse↔Zebra. In the experiments, our network architecture consists of five scales, represented from Scale $0 - 4$ respectively. In the later experimental results, Scale $0 - M$ represents the current block trained from Scale 0 to Scale $M$. The weights obtained from Scale $M$ training are directly applied to Scale $M + 1$. Since the Tail block in our network architecture directly outputs the image translation result, consists of one layer of convolution and an activation function, so its training is required in all scales by default.

We design three parameter study experiments, Experiment 1 is used to verify the effect of the scaling factor $\eta$ on our model, and Experiments 2 and 3 to observe the effect of the parallel training mode on the model for the Head and Body blocks.

In Experiment 1, we set the Head, Body, and Tail blocks to participate in training at all Scale by default, vary the value of the scaling factor $\eta$, and observe the effect of different scaling factors $\eta$ on the model performance. To avoid overfitting in the lower scales due to too high

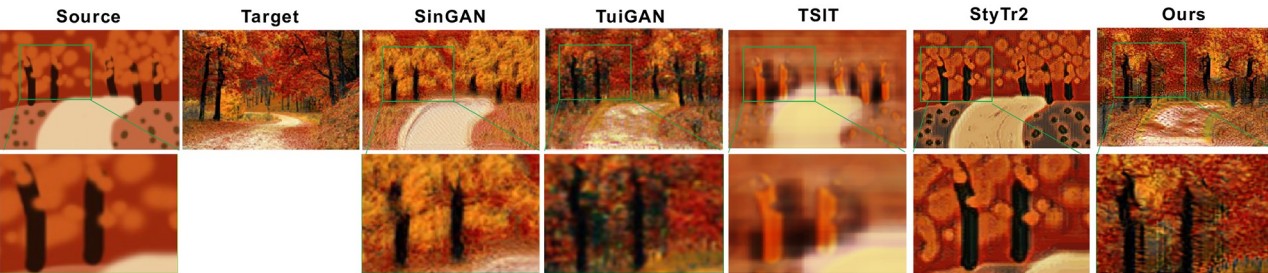

**Fig 9. Results of painting-to-image translation.** We amplify the green box in the translated image at the second row to show more detail.

| Source Image | Target Image | 0.05 | 0.10 | 0.30 | 0.50 | 1.0 |

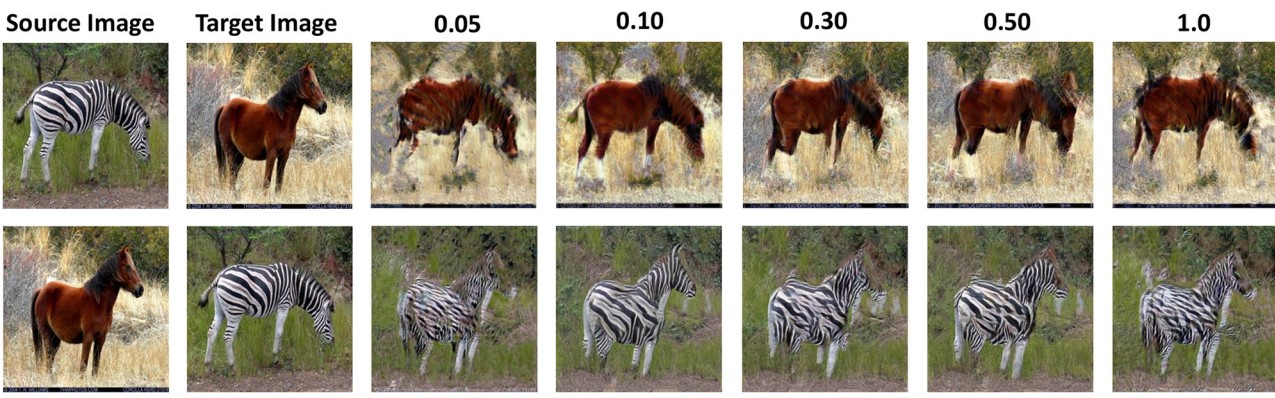

**Fig 10. Visual results of scale factor $\eta$.**

learning rate, we defaulte Head, Body, Tail blocks to be trained in all Scale stages, mitigate by scaling the learning rate and the factor $\eta$. We change the values of the scaling factor $\eta$ to 0.05, 0.10, 0.30, 0.50, 1.0 to observe the influence of $\eta$ to translate, and the results are shown in Fig 10 and Table 3. From the results, it can be seen that when $\eta$ is 0.05, the generator is unable to learn the local features of the target image and can only generate the outline of the horse, but not the specific texture. When the $\eta$ is 0.30, 0.50 or 1.0, the horse's head is distorted and texture is missing, whose reason is to the occurrence of overfitting. When $\eta$ is 0.10, the image translation results are better and the SIFID value is minimal, generating a more realistic image. Therefore, in our model CRPGAN, we fix the scaling factor $\eta$ to 0.10.

In Experiment 2, We fixed scaling factor $\eta$ to 0.10 and Body, Tail blocks to be trained in all Scale and varied the parallel training positions of Head blocks at different scales. The results of Experiment 2 are shown in Fig 11(a) and Table 4. From Fig 11(a), we can see that Scale 0 and Scale 0 − 1 cannot guarantee the object integrity (the translated zebra's head is missing), and Scale 0 − 2 appears to have incorrect color and texture (the translated zebra's head appears to have a horse texture). Scale 0 − 4 can capture the content features of the image and transfer the style features of the target image (the translated zebra with a horse). The Table 4 also shows that Scale 0 − 4 has the smallest SIFID value, representing a more realistic generated image. As the Scale of parallel training increase, the image content feature is better maintained and target image style is better transferred. Therefore, when the scaling factor $\eta$ is 0.10 and Body and Tail blocks are trained in all Scale, Head block trained in all Scale can ensure object integrity and translation accuracy.

Similar to Experiment 2, we also fixed the scaling factor $\eta$ to 0.10 in Experiment 3 and trained Head and Tail blocks at all Scale by default, varying the parallel training positions of Body blocks at different scales. From Fig 11(b), we can see that Scale 0 and Scale 0 − 1 cannot guarantee the integrity of the object (the translated zebra's head is missing), and as the Scale increases, Scale 0 − 3 and Scale 0 − 4 show incorrect colors and textures (the translated zebra's head shows a horse texture). Scale 0 − 2 captures the image content features, and transfer the

**Table 3. Quantitative comparison between different scale factor $\eta$ of CRPGAN in terms of SIFID.** The best scores are in bold.

| Metrics | CRPGAN (SIFID↓) | | | | |
|---|---|---|---|---|---|
| | **0.05** | **0.10** | **0.30** | **0.50** | **1.0** |
| **Horse→Zebra** | 0.105 | **0.071** | 0.081 | 0.075 | 0.096 |
| **Zebra→Horse** | 0.118 | **0.076** | 0.094 | 0.091 | 0.108 |

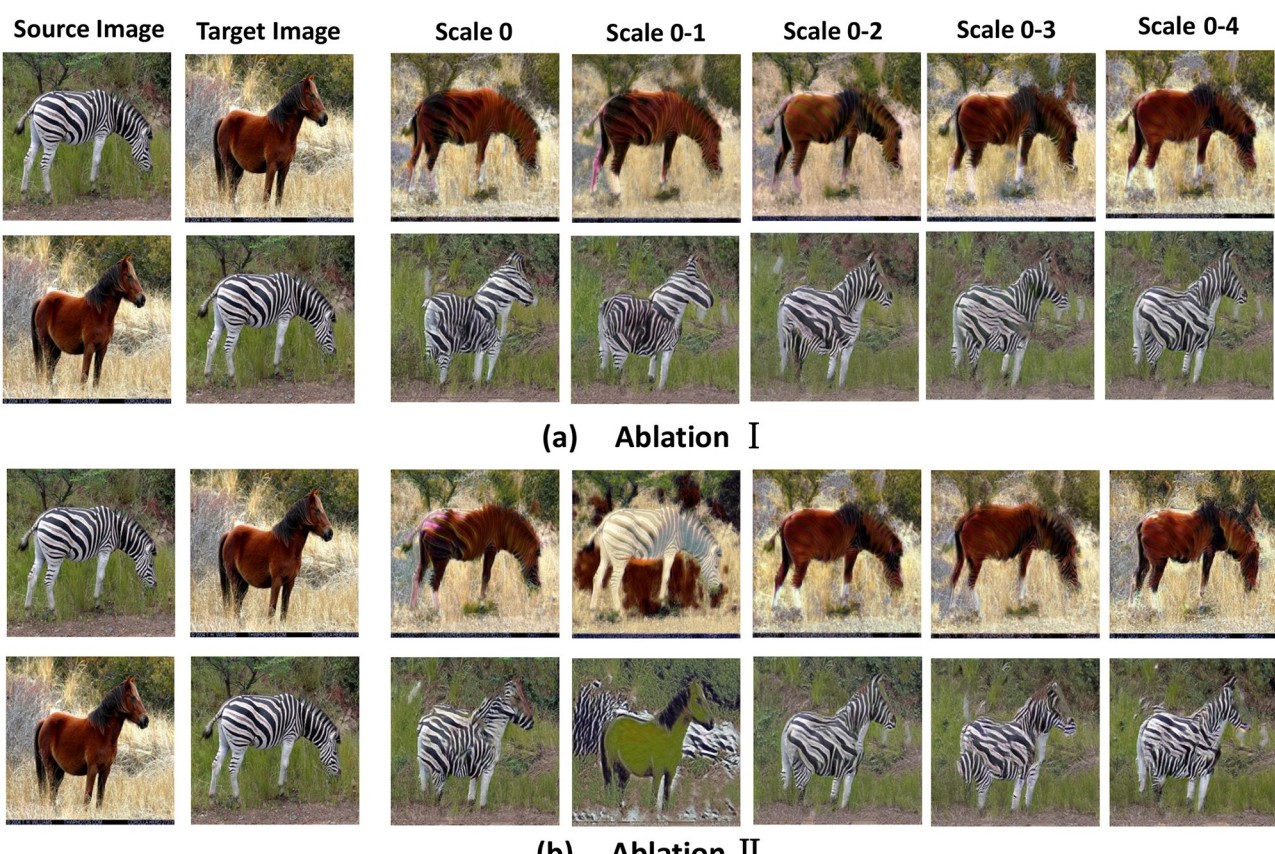

**Fig 11. Visual results of parametric study.**

style features (translated zebra with a horse style). This is also verified by the experimental results in Table 3. Combining Fig 11(b) and Table 3, it can be seen that when the scaling factor $\eta$ is 0.10 and Head and Tail blocks are trained at all Scale, Body block stops training at Scale 3, which can ensure the object integrity and translation accuracy. The reason is that the Body block learns global information of the image at Scale $0-2$ to ensure the integrity of the image, but over-training the Body block will result in overfitting and cause the image translation results to appear as features of the source image.

## D, Ablation study

To evaluate the effect of individual loss functions and *CRCBAM* attention in the CRPGAN on the image translation results, we design ablation experiments based on Summer↔Winter, as shown in Fig 12.

**Table 4. Quantitative comparison between different scale factor $\eta$ of CRPGAN in terms of SIFID.** The best scores are in bold.

| Ablation | Metrics | CRPGAN (SIFID↓) | | | | |
|---|---|---|---|---|---|---|
| | | Scale 0 | Scale 0-1 | Scale 0-2 | Scale 0-3 | Scale 0-4 |
| I | Horse→Zebra | 0.098 | 0.075 | 0.081 | 0.076 | **0.068** |
| | Zebra→Horse | 0.107 | 0.101 | 0.091 | 0.088 | **0.071** |
| II | Horse→Zebra | 0.089 | 0.109 | **0.068** | 0.093 | 0.082 |
| | Zebra→Horse | 0.095 | 0.117 | **0.071** | 0.099 | 0.088 |

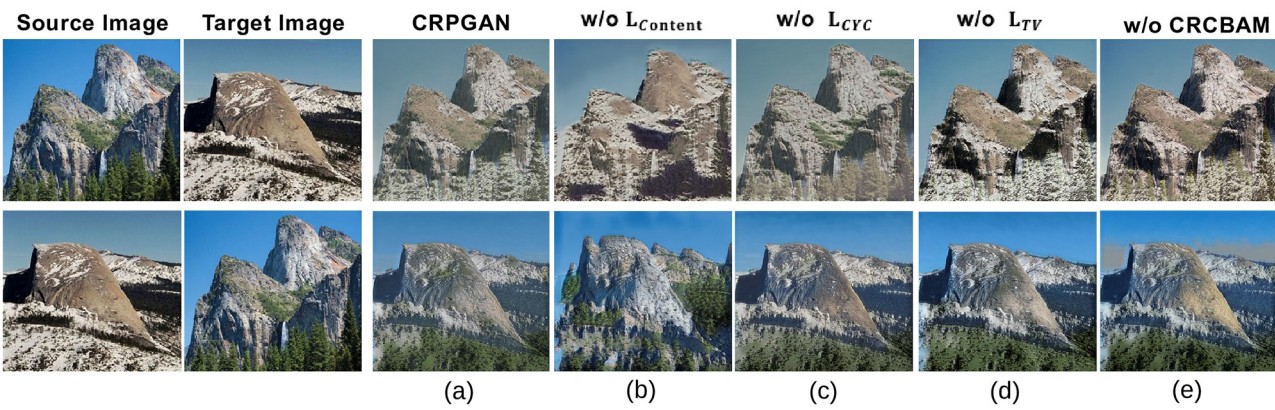

**Fig 12. Visual results of ablation study.**

**Table 5. Quantitative comparisons results for CRPGAN ablations study in terms of SIFID.** The best scores are in bold.

| Metrics | CRPGAN (SIFID↓) | | | | |
|---|---|---|---|---|---|
| | CRPGAN | w/o $L_{Content}$ | w/o $L_{CYC}$ | w/o $L_{TV}$ | w/o *CRCBAM* |
| **Summer→Winter** | **0.385** | 1.597 | 1.255 | 1.273 | 1.155 |
| **Winter→Summer** | **0.496** | 1.619 | 1.080 | 1.047 | 1.290 |

Fixing N = 6, epochs = 100, we removed content loss (CRPGAN w/o $L_{Content}$), cyclic consistency loss (CRPGAN w/o $L_{CYC}$), total variation loss(CRPGAN w/o $L_{TV}$), *CRCBAM* attention mechanism(CRPGAN w/o *CRCBAM*) and compared the differences.

On image style transfer tasks such as Summer↔Winter, the qualitative result is shown in Fig 12 and the quantitative result is shown in Table 5. Without content loss $L_{Content}$, the content information of the generated image is lost (Fig 12(b)). Without cyclic consistency loss $L_{CYC}$, the generated result style information is lost(Fig 12(c)). Without total variation loss $L_{TV}$, Our model generates image with noise(Fig 12(d)). Without *CRCBAM*, our model also loses location information (Fig 12(e)). Our method generates finer stylized images while preserving the source image content with the lowest SIFID value(Fig 12(a)).

## E, Object transformation

To verify the generalizability of our method on other datasets, we conducte experiments on the animal and OperaFace datasets.

In addition, we show the results of CRPGAN on four image object transformation tasks, which are dog face Translation, cat face translation, wild face translation, and OperaFace translation in Fig 13. The experiment results verify the the generality of the model in this paper on the UI2I tasks with good performance, our model can generate more realistic and higher quality translated images.

## Conclusions and discussion

In this paper, we propose CRPGAN, a new image-to-image translation framework with two unpaired images. Specifically, CRPGAN uses a multi-scale training process to learn the global and local structures (texture and style features) of images from coarse to fine. Meanwhile, we use a progressive growth generator to grow the generator size at each scale and adjust the

**Source Image    Target Image    Translated Image  Source Image    Target Image    Translated Image**

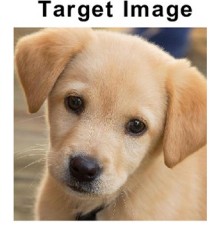 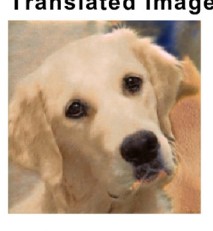 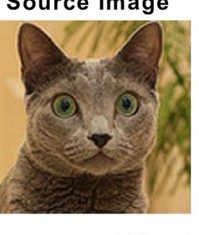 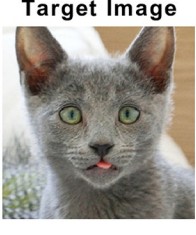 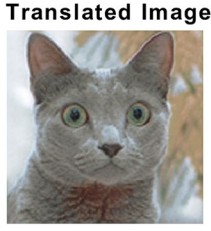

(a) Dog  Face Translation                    (b) Cat  Face Translation

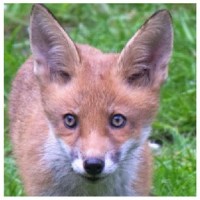 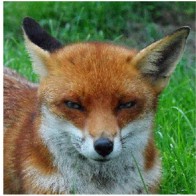 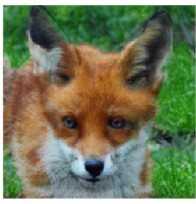 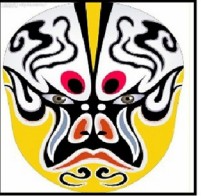 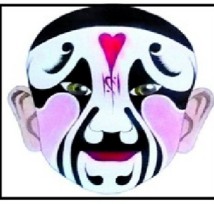 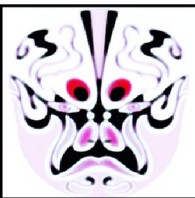

(c) Wild  Face Translation                    (d) Opera  Face Translation

**Fig 13. Our model can accurately transfer animal hair color and opera face style features.**

learning rate at lower scales and the number of layers in parallel training stages so that the model can accurately capture the differences in the distribution between the source and target domains and improve the quality of translated images. Next, the model training speed is improved by using a twice-parameter sharing structure. Finally, the newly proposed CRCBAM can fully extract the local and global information of single-sample images to generate finer stylized images. The experimental results show that in image translation tasks with extremely limited data, our method can make better use of image information to generate detailed and realistic image translation results, and the framework can be widely applied to image translation tasks.

However, there are still shortcomings in our method, the first is the generalization problem, such as the use of horses and zebras trained models to use the same horse dataset to translate poorly. We consider the possibility of introducing data augmentation to improve the performance in the future. Secondly, the incomplete parallel Strategy. The parallel Strategy does not need to repeatedly train the weights of the same Body layer in the current stage, but still needs to utilize the training weights of the previous stage, and it is future to study the real sense of parallel strategies in the training process of single-sample image translation. Finally, transformer [54] can learn image global information better, which can guarantee the original information in the image style transfer and translation, but it needs to be trained on large datasets, and how to use transformer in single-sample image translation is a scientific research point to explore.

## Author Contributions

**Conceptualization:** Kang Li.

**Data curation:** Qihang Li.

**Formal analysis:** Long Feng, Guohua Geng, Zhan Li.

**Funding acquisition:** Guohua Geng, Kang Li.

**Investigation:** Guohua Geng, Qihang Li, Zhan Li.

**Methodology:** Long Feng, Guohua Geng, Zhan Li.

**Software:** Long Feng, Qihang Li, Yi Jiang.

**Writing – original draft:** Long Feng, Kang Li.

**Writing – review & editing:** Guohua Geng, Zhan Li.

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
