## [Decision Letter · Decision Letter 0]

6 Nov 2022

PONE-D-22-26766CRPGAN: Learning image-to-image translation of two unpaired images by cross-attention mechanism and parallelization strategyPLOS ONE

Dear Dr. Geng,

Thank you for submitting your manuscript to PLOS ONE. After careful consideration, we feel that it has merit but does not fully meet PLOS ONE’s publication criteria as it currently stands. Therefore, we invite you to submit a revised version of the manuscript that addresses the points raised during the review process.

We look forward to receiving your revised manuscript.

Kind regards,

Xiangjie Kong

Academic Editor

PLOS ONE

Journal Requirements:

"This research was funded by the National Key Research and Development Program of China (2020YFC1523301 and 2019YFC1521103), Key Research and Development Program of Shaanxi Province (2019ZDLSF07-02, 2019ZDLGY10-01 and 2021GY-171), National Natural Science Foundation of China (61731015), Key Research and Development Program of Qinghai Province (2020-SF-142)."

"The authors received no specific funding for this work."

4. We note that Figures 1,2,5-14 in your submission contain copyrighted images. All PLOS content is published under the Creative Commons Attribution License (CC BY 4.0), which means that the manuscript, images, and Supporting Information files will be freely available online, and any third party is permitted to access, download, copy, distribute, and use these materials in any way, even commercially, with proper attribution. For more information, see our copyright guidelines: http://journals.plos.org/plosone/s/licenses-and-copyright.

a) You may seek permission from the original copyright holder of Figures 1,2,5-14 to publish the content specifically under the CC BY 4.0 license. 

5. Please ensure that you refer to Figure 4 in your text as, if accepted, production will need this reference to link the reader to the figure.

Additional Editor Comments:

Please revise paper according to the comments and suggestions from the reviewers.

Reviewers' comments:

Reviewer's Responses to Questions

**Comments to the Author**

1. Is the manuscript technically sound, and do the data support the conclusions?

Reviewer #1: Yes

Reviewer #2: Yes

2. Has the statistical analysis been performed appropriately and rigorously? 

Reviewer #1: Yes

Reviewer #2: Yes

3. Have the authors made all data underlying the findings in their manuscript fully available?

Reviewer #1: Yes

Reviewer #2: Yes

4. Is the manuscript presented in an intelligible fashion and written in standard English?

Reviewer #1: Yes

Reviewer #2: Yes

5. Review Comments to the Author

Reviewer #1: This manuscript introduced the cross-attention mechanism and parallelization strategy into GAN for image-to-image translation of unpaired images. The idea is interesting and the result is basically satisfactory.

Reviewer #2: This paper proposes a multi-scale training structure and a progressive growth generator method for unsupervised image-to-image translation tasks. My concerns are as follows:

1. The proposed method refines the generated images by adding new convolution blocks continuously in different scales, and generates more refined style images with a new Cross-CBAM attention mechanism.

2. The effectiveness of the proposed method should be further verified. In the original paper of comparison methods TuiGAN and SinGAN, the SIFID results are averaged across different general UI2I tasks, while comparison experiments are only conducted on 2 tasks (Summer-Winter, Horse-Zebra). The other datasets are not mentioned in the paper, such as OperaFace, Monet-Photo, GrumpifyCat, and Van Gogh-Photo. The settings of experiments should be identical for a fair comparison.

3. The latest comparison methods are published in 2020, and more recent SOTA methods, such as [a], [b], should be compared to further verify the effectiveness of the proposed method.

4. There are many typos in the paper. For example, the number of references for method TSIT in table 2 should be [44], instead of [25]. And the name of the proposed method happens to be “ParGAN” in the last second paragraph of the Introduction, which should be “CRPGAN”. The authors may want to proofread the whole paper to carefully polish the writing.

5. The visualization results should be further refined. For example, the difference between columns (b), (c), and (d) is not clear enough to directly show the advantages of the proposed Cross-CBAM attention mechanism.

6. The end of each line should be aligned for a better outlook of the paper.

7. More related works about “GAN”’s application on different tasks should be included to improve the completeness of the paper, such as [c], [d], [e], and [f].

[a]: Dual Contrastive Learning for Unsupervised Image-to-Image Translation, CVPR 2021

[b]: Unaligned Image-to-Image Translation by Learning to Reweight, ICCV 2021

[c]: Multi-View Consistent Generative Adversarial Networks for 3D-aware Image Synthesis, CVPR 2022

[d]: Dual Projection Generative Adversarial Networks for Conditional Image Generation, ICCV 2021

[e]: Bridge-GAN: Interpretable Representation Learning for Text-to-image Synthesis, TCSVT 2020

[f]: CM-GANs: Cross-modal Generative Adversarial Networks for Common Representation Learning, TOMM 2019

6. PLOS authors have the option to publish the peer review history of their article (what does this mean?). If published, this will include your full peer review and any attached files.

Reviewer #1: No

Reviewer #2: No

---

## [Author Response · Author response to Decision Letter 0]

18 Nov 2022

Dear Editors and Reviewers:

Thank you for your letter and for the reviewers’comments concerning our manuscript entitled “CRPGAN: Learning image-to-image translation of two unpaired images by cross-attention mechanism and parallelization strategy” (ID: PONE-D-22-26766). Those comments are all valuable and very helpful for revising and improving our paper, as well as the important guiding significance to our researches. We have studied comments carefully and have made correction which we hope meet with approval. Revised portion aremarked in blue in the paper. The main corrections in the paper andthe responds to the reviewer’s comments are as flowing:

Responds to the reviewer’s comments:

Reviewer #1:

Response to comment: In the experiments, the proposed method was compared with four methods, which are published in 2017, 2018, 2019, and 2020. In order to validate the effectivity more extensively, could the authors compare with more recent works?

Response: We added the image translation methods of the last two years as a baseline for qualitative and quantitative experiments with the methods in this paper, mainly DCLGAN, lrwGAN in 2021 and StyTr2, Qs-Attn in 2022. 

DCLGAN bases on contrastive learning and a dual learning setting to infer an efficient mapping between unpaired data. lrwGAN solves the image translation task between two unaligned domains by importance re-weighted image selection. StyTr2 is a style transfer model using transformers as encoders. Qs-Attn design a query-selected attention (QS-Attn) module to ensure that the source image learns the target image features at the corresponding location for image translation. All of the above methods are typical of image translation and are similar to the data used in our method. 

DCLGAN in our reference of [19]. lrwGAN in our reference of [23]. StyTr2 in our reference of [53]. Qs-Attn in our reference of [24].

Response to comment: Could the authors do ablation experiments on loss function?

Response: The loss function of the ablation experiment conducted in this paper is in Experiment in D, Ablation Study on the page 17. 

We designed ablation experiments based on Summer↔Winter in order to verify the role of content loss, cyclic consistency loss, total variation loss and Cross-CBAM mechanism in one-shot image translation. We found that content loss preserves image content information, cyclic consistency loss prevents loss of image information, total variation loss prevents noise in images, and Cross-CBAM mechanism is used to generate finer style images.

Response to comment: The keywords are missing?

Response: We are very sorry that the Latex template for PLOS One cannot add keywords. But we have keywords, which are Unsupervised image-to-image translation, multi-scale training, progressive growth generator, parameter sharing, CRCBAM mechanism.

Response to comment:The organization of this manuscript should be added to the end of the introduction.

Response: We have added the organization of the article and located it at the end of the introduction, in the red section on the page 4 . The organization of this manuscript is as follows:

The paper is structured as follows. Section 1 is a general introduction providing the necessary background and problems of unsupervised image translation. Section 2 provides current work related to style transfer, unsupervised image translation, and one-shot image translation. Section 3 provides the network structure, attention mechanism and loss function of the methods in this paper. Section 4 provides experimental setup, baseline methodology and evaluation metrics. Section 5 provides qualitative and quantitative experiments and the corresponding parametric and ablation studies. Section 6 provides a conclusion of our approach and future work.

Response to comment:More recent works on image translation could be included.

Response: We have added new image translation jobs, The references are as follows

[13]: Multi-View Consistent Generative Adversarial Networks for 3D-aware Image Synthesis, CVPR 2022

[14]: Bridge-GAN: Interpretable Representation Learning for Text-to-image Synthesis, TCSVT 2020

[15]: Dual Projection Generative Adversarial Networks for Conditional Image Generation, ICCV 2021

[16]: CM-GANs: Cross-modal Generative Adversarial Networks for Common Representation Learning, TOMM 2019

[19]: Dual Contrastive Learning for Unsupervised Image-to-Image Translation, CVPR 2021

[21]: Asynchronous generative adversarial network for asymmetric unpaired image-to-image translation. IEEE Transactions on Multimedia, 2022.

[22]: A deep translation (GAN) based change detection network for optical and SAR remote sensing images. ISPRS Journal of Photogrammetry and Remote Sensing, 2021

[23]: Unaligned Image-to-Image Translation by Learning to Reweight, ICCV 2021

[24]:QS-Attn: Query-Selected Attention for Contrastive Learning in I2I Translation, CVPR 2022

[53]: StyTr2: Image Style Transfer with Transformers, CVPR 2022

Response to comment: Some references missed the fundamental information.

Response: Thank you very much for your correction. We have carefully checked each referenceand revised it as follows: for the references cited with DOI, the URL prefix was removed and only the DOI number was retained.

Special thanks to you for your good comments.

Reviewer #2:

Response to comment: The proposed method refines the generated images by adding new convolution blocks continuously in different scales, and generates more refined style images with a new Cross-CBAM attention mechanism.

Response: Thank you for your affirmation and other very constructive suggestions. We will continue our efforts to delve deeper into issues in other areas.

Response to comment: The effectiveness of the proposed method should be further verified. In the original paper of comparison methods TuiGAN and SinGAN, the SIFID results are averaged across different general UI2I tasks, while comparison experiments are only conducted on 2 tasks (Summer-Winter, Horse-Zebra). The other datasets are not mentioned in the paper, such as OperaFace, Monet-Photo, GrumpifyCat, and Van Gogh-Photo. The settings of experiments should be identical for a fair comparison.

Response: Considering your suggestion, we have added photo→Van Gogh for qualitative and quantitative experiments of the baseline method. However, we did not add this experiment again due to the similarity between photo→Monet and photo→Van Gogh. The image translation between horse and zebra can represent the image translation between objects, therefore, in the object translation methods, such as OperaFace and GrumpifyCat, we did not add the remaining sets of comparison experiments, and only put the experimental results of the methods in this paper in the E, Object Transformation section of the final Result. Thank you again for your guidance and for making us understand the benefits of comparative experiments.

Response to comment: The latest comparison methods are published in 2020, and more recent SOTA methods, such as [a], [b], should be compared to further verify the effectiveness of the proposed method.

Response: We added the image translation methods of the last two years as a baseline for qualitative and quantitative experiments with the methods in this paper to further verify the effectiveness of the proposed method, mainly DCLGAN[a], lrwGAN[b] in 2021 and StyTr2, Qs-Attn in 2022.

DCLGAN bases on contrastive learning and a dual learning setting to infer an efficient mapping between unpaired data. lrwGAN solves the image translation task between two unaligned domains by importance re-weighted image selection. StyTr2 is a style transfer model using transformers as encoders. Qs-Attn design a query-selected attention (QS-Attn) module to ensure that the source image learns the target image features at the corresponding location for image translation. All of the above methods are typical of image translation and are similar to the data used in our method. 

DCLGAN in our reference of [19]. lrwGAN in our reference of [23]. StyTr2 in our reference of [53]. Qs-Attn in our reference of [24].

Response to comment: There are many typos in the paper. For example, the number of references for method TSIT in table 2 should be [44], instead of [25]. And the name of the proposed method happens to be “ParGAN” in the last second paragraph of the Introduction, which should be “CRPGAN”. The authors may want to proofread the whole paper to carefully polish the writing.

Response: Thank you very much for your correction. We double-checked the index of the paper and the wrong words, and have revised the problems. Revise the following: correct the incorrect "ParGAN" to "CRPGAN", and correct the incorrect reference number [25] in TSIT to [52] after adding the reference.

. 

Response to comment:The visualization results should be further refined. For example, the difference between columns (b), (c), and (d) is not clear enough to directly show the advantages of the proposed Cross-CBAM attention mechanism.

Response: Considering your suggestion, in Figure 5(a)(b)(c)(d) on the page 10, we have partially scaled up the experimental results of the Cross-CBAM attention mechanism, and we can find that the method(d) in this paper preserves more information. The translated images in b and c relatively blur at the edges and generate a large amount of erroneous blue information on the left side.

Response to comment: The end of each line should be aligned for a better outlook of the paper.

Response: Thank you for your corrections on the typography issue. We are very sorry that this is a requirement of the PLos One template. 

Response to comment: More related works about “GAN”’s application on different tasks should be included to improve the completeness of the paper, such as [c], [d], [e], and [f].

Response: We have added new image translation jobs, such as [13], [14], [15], [16], [19], [21], [22], [23], [24] and [53] and used [15], [23], [24] and [53] as baseline methods.

[13]: Multi-View Consistent Generative Adversarial Networks for 3D-aware Image Synthesis, CVPR 2022

[14]: Bridge-GAN: Interpretable Representation Learning for Text-to-image Synthesis, TCSVT 2020

[15]: Dual Projection Generative Adversarial Networks for Conditional Image Generation, ICCV 2021

[16]: CM-GANs: Cross-modal Generative Adversarial Networks for Common Representation Learning, TOMM 2019

[19]: Dual Contrastive Learning for Unsupervised Image-to-Image Translation, CVPR 2021

[21]: Asynchronous generative adversarial network for asymmetric unpaired image-to-image translation. IEEE Transactions on Multimedia, 2022.

[22]: A deep translation (GAN) based change detection network for optical and SAR remote sensing images. ISPRS Journal of Photogrammetry and Remote Sensing, 2021

[23]: Unaligned Image-to-Image Translation by Learning to Reweight, ICCV 2021

[24]:QS-Attn: Query-Selected Attention for Contrastive Learning in I2I Translation, CVPR 2022

[53]: StyTr2: Image Style Transfer with Transformers, CVPR 2022

Special thanks to you for your good comments.

Other changes:

1.Removed the "B, Dataset in Experiment"

We tried our best to improve the manuscript and made some changes in the manuscript. These changes will not influence the content and framework of the paper. And here we did not list the changes but marked in red in revised paper.

We appreciate for Editors/Reviewers’ warm work earnestly, and hope that the correction will meet with approval.

Once again, thank you very much for your comments and suggestions.

---

## [Decision Letter · Decision Letter 1]

20 Dec 2022

CRPGAN: Learning image-to-image translation of two unpaired images by cross-attention mechanism and parallelization strategy

PONE-D-22-26766R1

Dear Dr. Geng,

We’re pleased to inform you that your manuscript has been judged scientifically suitable for publication and will be formally accepted for publication once it meets all outstanding technical requirements.

Kind regards,

Xiangjie Kong

Academic Editor

PLOS ONE

Additional Editor Comments (optional):

Reviewers' comments:

Reviewer's Responses to Questions

**Comments to the Author**

1. If the authors have adequately addressed your comments raised in a previous round of review and you feel that this manuscript is now acceptable for publication, you may indicate that here to bypass the “Comments to the Author” section, enter your conflict of interest statement in the “Confidential to Editor” section, and submit your "Accept" recommendation.

Reviewer #1: All comments have been addressed

2. Is the manuscript technically sound, and do the data support the conclusions?

Reviewer #1: Yes

3. Has the statistical analysis been performed appropriately and rigorously? 

Reviewer #1: Yes

4. Have the authors made all data underlying the findings in their manuscript fully available?

Reviewer #1: Yes

5. Is the manuscript presented in an intelligible fashion and written in standard English?

Reviewer #1: Yes

6. Review Comments to the Author

Reviewer #1: The authors have answered all of my concerns in the last version. The current version can be published.

7. PLOS authors have the option to publish the peer review history of their article (what does this mean?). If published, this will include your full peer review and any attached files.

Reviewer #1: No

---

## [Editor Report · Acceptance letter]

27 Dec 2022

PONE-D-22-26766R1 

CRPGAN: Learning image-to-image translation of two unpaired images by cross-attention mechanism and parallelization strategy 

Dear Dr. Geng:

I'm pleased to inform you that your manuscript has been deemed suitable for publication in PLOS ONE. Congratulations! Your manuscript is now with our production department. 

Kind regards, 

on behalf of

Dr. Xiangjie Kong 

Academic Editor

PLOS ONE